# Temporal Dimensions of Job Quality and Gender: Exploring Differences in the Associations of Working Time and Health between Women and Men

**DOI:** 10.3390/ijerph19084456

**Published:** 2022-04-07

**Authors:** Paula Franklin, Wouter Zwysen, Agnieszka Piasna

**Affiliations:** European Trade Union Institute (ETUI), 1210 Brussels, Belgium; wzwysen@etui.org (W.Z.); apiasna@etui.org (A.P.)

**Keywords:** job quality, working time, gender, women’s health, health inequalities

## Abstract

Research shows strong links between working time organization and workers’ health outcomes. Working time is also known to be highly gendered, with men and women working to different schedules. This article merges these two strands of research and takes a gender-based approach to investigating the relationship between temporal job quality and self-reported health in Europe. First, the sixth European Working Conditions Survey (EWCS) is used to establish the relationship between temporal dimensions of job quality and health and well-being outcomes for employed women and men. This is then corroborated using larger samples and more restricted measures of job quality drawn from micro-data from the 2019–2020 EU Labor Force Survey (LFS). The analyses show that good temporal job quality is positively associated with health and subjective well-being for both women and men, but this effect is significantly stronger for women, who are also at a greater risk of exposure to low control over working time and time under-employment. The findings highlight the importance of studying the impact of working and employment conditions on health from gender perspective, and the need for further exploration of job quality due to changes in the spatio-temporal organization of work during and beyond the COVID-19 pandemic.

## 1. Introduction

Work is an important determinant of health and gendered employment patterns are reflected in health inequities. Vertical and horizontal segregation means that women and men occupy different positions in the labor market, with different working and employment conditions and subsequent health effects [1,2,3]. Time is inherent to the experience of work and the way it is organized with gender being a defining factor in this context [4,5,6]. Men have traditionally been more exposed to negative health effects from long-work hours, and women are more likely to be underemployed and experience physical and mental health problems as a result of work-family strain from the dual burden of employment and caring [7]. Across the whole European Union (EU), women continue to work according to different schedules than men, a fact predominantly related to the unequal division of household and care work [8].

In this article, we explore the relationship between job quality and self-reported health, with a specific focus on working time. We will first discuss the evidence on working time impact on health, followed by an analysis of European data on the relationship between temporal dimensions of job quality, gender, and work having negative effect on health. The study contributes to research on gender inequalities in health and their social determinants through a gender-sensitive analysis of a link between health and indicators related to working conditions, in particular temporal job quality.

Time is both a quantitative measure and a qualitative dimension of work and a key element in examining job quality and health from a gender perspective [9]. Long daily hours tend to be associated with acute effects of fatigue, and long weekly hours both with acute effects of fatigue as well as chronic fatigue, and generating long-term negative health effects, including burnout, occupational stress, depression, anxiety, and other mental health disorders [10]. The risk of ischemic heart disease and stroke are attributed to working more than 55 h a week [11], although it is suggested that this applies only to workers with a lower socio-economic status [12].

A recent review by Moreno et al. [13] of the epidemiological evidence on the association between shiftwork and health concluded that there is a strong link between shiftwork and negative physical health outcomes, such as cardiovascular diseases, gastrointestinal and metabolic disorders (e.g., type 2 diabetes). The evidence also associates shiftwork to cancer and reproduction related problems. Research also shows that poor mental health can be a consequence of shift work [14]. Moreno et al. [13] note that men and women can respond differently to shiftwork, and therefore the risk of developing negative health effects may be gender specific.

Work-life conflict (WLC) among employees is known to be related to many health problems, including poor physical health, poor-self reported health, and poor mental health [15,16,17]. Time-based work-to-life conflict, specifically, has been found to have strong association with burnout [18] and in contributing to musculoskeletal problems [19]. Henly and Lambert’s (2014) study found unpredictable work timing in retail jobs positively associated with time-based conflict as measured by perceived employee stress [20]. WLC and health outcomes may differ by gender due to the unequal distribution of work-related roles, but results for gender-specific health outcomes remain inconclusive; some studies have found a positive link between WLC and poor self-reported health of working women but not men [21,22], and Hegewald et al. (2021) found that WLC may be negatively impacting the cardiovascular health of women [23]. Other studies suggest similar outcomes between men and women [24].

Moen et al.’s (2013) study of the relationship between work time demands and control and self-reported health showed that psychological time demands, and time control measures are related to health outcomes [25]. The European survey on new and emerging risks (ESENER) on psychosocial risk factors (PSR) reported by European workplaces found that the second most reported risk is ‘pressure due to time’ [26]. Flexibility and control of employees over their working time—both task-related pressure and the overall structuring of working time—have been associated with positive health outcomes [10,27,28,29].

Epidemiological studies have established that precarious or temporary work is associated with adverse health outcomes as a result of low control over working hours and work-life conflict, among other psychosocial risk factors (PSR) [30,31,32,33]. Research in the US, UK, and Finland shows that under-employment—i.e., working fewer hours than one would prefer and involuntary part-time work, negatively affects employees’ mental health and well-being, especially in women [34,35,36,37]. A study in Denmark showed poorer health of marginal part-time workers (8.0–14.9 h/week) as compared to full-time workers (32.0–40.0 h/week), mediated by poor working conditions and job insecurity, women being in the majority of this worker segment [38]. Evidence on positive health impacts of shorter working hours highlights the importance of job quality (e.g., work needs to be intrinsically meaningful, less intense, and have a favourable social environment) [39,40,41]. In general, part-time employment is associated with poorer working conditions than full-time employment across the EU [42].

As outlined above, research shows strong links between working time and workers’ health outcomes. Several different aspects of working time have been investigated in relation to health, including duration of working hours (daily and weekly) and working time arrangements (the ways in which the working hours are organized). Further, the COVID-19 pandemic has caused an abrupt change in the spatio-temporal organization of work, due to compulsory telework in many countries, and school and daycare closures that increased caregiving responsibilities for working parents. The possibly permanent shift to more hybrid and remote working modes makes temporal analysis of job quality highly relevant. It is within this background that we analyse how temporal dimensions of job quality mediate gendered health impacts.

## 2. Materials and Methods

The objective of this article is to analyse gender differences in the impact of temporal dimensions of job quality on workers’ health and well-being. To establish the relationship between temporal dimensions of job quality and health and well-being outcomes for workers we use two large cross-national representative European datasets.


*Study 1: Detailed measures of job quality and health outcomes based on the EWCS data*


First, we use data from the sixth European Working Conditions Survey (EWCS) carried out by Eurofound in 2015 to establish the relation between detailed aspects of temporal dimensions of job-quality and health. The EWCS microdata were obtained through the UK Data Service (UKDS) in Essex. The EWCS is a cross-sectional survey with a usual sample size of 1000 per country representative of persons in employment (for a detailed description of survey methodology see Eurofound 2016) [43]. The analysis includes a sample of 35,765 workers from 28 European countries (EU-27 and the UK).

Job quality is a multi-dimensional concept and in the literature is measured on several dimensions, which characterise features of jobs linked to positive outcomes for workers [44,45]. We focus the analysis on the temporal dimension of job quality, which is related to the duration and organization of working hours, as well as time pressure, an element of work intensity. Drawing on previous studies that also use the EWCS to construct job quality indicators [8,43,46], we construct four indicators of specific elements of temporal job quality and one summary index that combines all of these four indicators. Table 1 provides a detailed description of job quality indicators used in the analysis: (JQ1) unsocial hours; (JQ2) long hours; (JQ3) flexibility and control; (JQ4) time pressure; and (JQ) full index of temporal job quality. No arbitrary weighting was introduced in the calculation of job quality indices (see Leschke and Watt, 2014) [47], with all survey items contributing equally to the final score within each index. All indicators of job quality used in the analysis were normalized to the 0–100 range (using the formula z_i_ = (x_i_ − min(x))/(max(x) − min(x)) × 100), with higher values always indicating better job quality (e.g., higher values on the ‘JQ1 unsocial hours’ indicate working fewer unsocial hours).

Table 2 presents the measures of health and well-being outcomes derived from the EWCS that are used in the analysis. Two items indicate negative health outcomes: a subjective assessment by survey respondents that their work affects their health mainly negatively and a number of self-reported health problems in the past 12 months. Two further items measure positive individual outcomes: sustainability of work measured by a perceived ability to work in the current or a similar job until the age of 60, and a five-item World Health Organization’s Well-Being Index (WHO-5), which is among the most widely used questionnaires assessing subjective psychological well-being.

The analysis aims at establishing gender-specific health and well-being outcomes (dependent variables) associated with temporal job quality (main predictors). We run logistic regression for binary outcomes (negative impact on health and sustainability of work) and linear regression for ordinal outcomes (number of health problems and subjective well-being), weighted by post-stratification and cross-national weights. To test whether job quality has different effect on health and well-being for men and women, each model includes an interaction term between gender and job quality. Two models are computed with each indicator of job quality for any of the four dependent variables: one model controlling for age group (under 35, 35–49, 50 and older), education (low, up to lower secondary; medium, upper secondary or post-secondary non-tertiary; high, tertiary), and country fixed effects, and a second model additionally controlling for occupational (nine groups based on 1-digit ISCO-08 classification, excluding armed forces) and sectoral (13 groups based on 1-digit NACE rev. 2 classification) composition. In total 40 regression models are computed in this step of the analysis. Descriptive statistics for all variables used in the models are provided in the Appendix A.


*Study 2: Health and Job Quality in the EU Labor Force Survey*


In addition to the EWCS data, we analyze gender differences in the temporal dimensions of job quality on micro-data from the EU Labor Force Surveys (LFS) made available by Eurostat. The analyses build on two ad-hoc modules: the 2019 module on work organization and working time arrangements; and the 2020 module on accidents at work and other work-related health problems. Data on all EU-27 countries, plus the United Kingdom and Norway, are used where available. The sample is restricted to those who work and are aged between 16 and 65. Table 3 describes the different variables used and how they are constructed. All variables are coded from 0 to 1, meaning continuous variables are recoded.

The LFS includes several indicators of the temporal dimension of job-quality but is generally far less detailed than the EWCS. The dimension of unsocial hours is captured by two indicator variables: one indicating workers generally work evenings, nights, or weekends; and one indicating they generally do shift work. To capture long hours an indicator for usually working more than 48 h per week is included. Two further variables are included to capture working time, namely whether respondents work part-time and whether they work part-time as they could not find a full-time position. The 2019 ad-hoc module includes more variables that can measure the other dimensions of job quality. Flexibility and control are approximated through two variables indicating the freedom to take leave or to take hours off; a variable indicating working time is decided on by the employer, as well as the extent to which working hours can be flexibly adjusted or whether workers have to be available. Finally, one variable captures the frequency of work under time pressure. All variables are coded from 0 to 1 with 1 indicating a more constrained position. Descriptive statistics for all variables used in the models are provided in the Appendix A.

Using the 2020 LFS ad-hoc module on health, we first reproduce the gender-specific analysis of a relationship between temporal job quality and health carried out with the EWCS data. This serves to corroborate our initial findings with larger sample sizes, yet narrower measures of job quality. We thus run a logistic regression of the LFS indicators (working part-time, working long hours, working on a temporary contract, working unsocial hours, and working in shifts) on four separate health outcomes (health problems, serious health problems, exposure to physical health risk, exposure to mental well-being risk factors). The analysis is weighted using the LFS-provided weights accounting for sampling probability and post-stratification weights. Each indicator of job quality is introduced in a separate regression and interacted with a gender dummy. Results are presented as the difference in the estimated effect of job quality on the probability of having experienced health problems of women from that of men. The models control for cohabiting, educational attainment, age, country fixed effects, occupational fixed effects, and industry fixed effects.

Finally, the paper establishes the extent of gender inequality in these different dimensions of temporal job quality using the 2019 LFS ad-hoc module. Two models are estimated for each of the indicators from the main LFS and the ad-hoc module: a first one controlling for basic socio-demographic information (cohabiting status, education, and age) as well as country fixed effects; and a second one including fixed effects for the combination of occupation and sector to capture the role sorting plays in gender differences in job quality.

## 3. Results

### 3.1. Study 1: Gender Differences in the Relationship between Temporal Dimensions of Job Quality and Workers’ Health and Well-Being

In general, workers with better quality jobs on any temporal dimension report less often that their job has a negative effect on their health and report positive health and well-being outcomes more often. Moreover, for women the effect of job quality is stronger than for men.

Figure 1 shows that better job quality on all analyzed temporal dimensions significantly (*p* < 0.01) correlates with a lower risk of a negative impact of work on health. It appears that the duration of working hours plays a smaller role, while time pressure in particular links to negative health outcomes. This means that, as shown in Figure 1 panel a, work in a high-quality job (the 90th percentile) compared to low-quality job (the 10th percentile) in terms of time pressure (JQ4) is associated with a lower risk of a negative impact of work on heath by 29% for men and by 32% for women. For women effects of all job quality dimensions are significantly stronger (*p* < 0.05) than for men (Figure 1a). However, when occupational and sectoral gender segregation is also accounted for (Figure 1b), gender differences are no longer significant for the effect of JQ1 unsocial hours and JQ2 long hours.

These findings are corroborated by the analysis of the association between job quality and health problems in the past 12 months (Figure 2), which shows a significant (*p* < 0.01) and positive relationship between job quality and better health outcomes. Moreover, we find that the effect is significantly stronger (*p* < 0.05) for women for JQ1 unsocial hours, JQ4 time pressure, and the overall index of job quality.

Temporal dimensions of job quality are positively associated with sustainability of work (Figure 3) and subjective well-being (Figure 4). The positive effect of job quality on sustainability of work is significantly stronger for women (*p* < 0.05) for all dimensions of job quality, except for JQ3 flexibility and control. Comparing workers with overall high job quality (90th percentile on JQ Full index) with those in low quality jobs (10th percentile), the probability of work being sustainable is higher by 8% for men and by 19% for women (Figure 3a). The effect of JQ2 long hours is only significant for women but not for men: women who work long hours are more at risk to report that their job is not sustainable, while there is no such clear association for men. However, there are no significant gender differences in the positive association between temporal dimensions of job quality and subjective well-being. Detailed results for all models are provided in the Appendix A.

### 3.2. Study 2: Supportive Evidence from the Labor Force Survey

The analysis of the LFS data largely confirms these findings of a stronger association between temporal job quality and health outcomes for women than men, as shown in Figure 5. Detailed regression results are shown in Appendix A. Working part-time is actually associated with a lower risk of health problems overall for women, but they are more exposed to physical risk factors than men when working part-time, especially in involuntary part-time work. Working long hours is associated with much worse health outcomes for both men and women, with a somewhat larger effect for women on the probability of experiencing serious health problems. Most striking are the differences between men and women in the effect of unsocial hours and shift work. Both aspects of low temporal job quality are associated with a higher risk of health problems and a greater exposure to physical and mental well-being risk factors, but these are especially prominent for women. This analysis then confirms the greater importance of temporal job quality for women’s well-being at work.

Gender-specific health outcomes linked to working time organization result not only from a stronger impact of temporal job quality on women’s health and well-being, as shown above, but stem also from gender divisions in working time patterns. There are sizeable gender differences in the quality of working time, as shown in Figure 6. Full regression results are shown in the Appendix A. While a non-negligible part of this is due to men and women working in different sectors and at different occupational levels, these gaps remain even when comparing people in similar jobs. Overall, women are almost 20%-points more likely to work part-time than men and are also more likely to work involuntarily part-time. Women are also clearly at a disadvantage when it comes to having control over their working time—they are more likely to report that their employers unilaterally set their working time, and that it is very difficult for them to take hours or days off on short notice. On the other hand, men are more likely to work long and unsocial hours compared to women. This difference is especially stark when comparing men and women in similar jobs. Men are also more likely to be expected to come to work at a very short notice, face unforeseen demands for changed working time and face severe time pressure. On the whole, then, this shows possibly greater pressures for adaptability for men, but less control and opportunity for worker-oriented flexibility for women.

## 4. Discussion

Our analyses of the two European datasets (EWCS and LFS) on temporal aspects of job quality (long hours, flexibility and control, time pressure, unsocial hours, shift work), and employment conditions (part-time voluntary/involuntary) show that their health impacts are gendered. This stems from two dynamics. First, women’s health and well-being are to a greater extent affected negatively by poor temporal job quality. Secondly, women are exposed to different aspects of poor working time quality than men, with a greater risk of low control and under-employment. We also found a significant impact of job quality in reducing negative health outcomes; for women this was particularly related to lower time pressure and not working unsocial hours.

From the quantitative stance, our analysis found that the impact of working long hours on self-reported health is negative for both women and men, but long hours are more likely to cause serious health problems for women than for men. Previous studies have found that health effects start at a lower working hours threshold for women than for men [48,49], or at a higher threshold [50]; and studies by Jeon et al. (2020) and Shin et al. (2021) into the association between working hours and self-rated health found that men working short hours and women working long hours were at risk of poor health, mediated by type of work and work schedule [51,52].

The analysis of the qualitative dimensions of working time that are related to poor self-reported health found that women have lower control over their working time. This concerns deciding the timing of working hours or days and having the flexibility to take hours off at short notice. Lack of time control and flexibility are recognised as psychosocial risks at work that can cause stress and result in negative physical and mental health outcomes [53,54]. Recent research found that in Europe, the proportion of depression attributable to job strain (combination of high psychological demands and low decision latitude) is 17%. [55]. The flexibility to take some hours off at a short notice for personal or family matters is also related to work-life balance/conflict, and as women continue to carry the main care responsibility across countries, the lack of temporal control can be considered to be a gendered work stress factor.

The analysis shows that time pressure is strongly linked to negative health outcomes for both women and men. While the data show that men are more likely to face severe time pressure at work, and are more likely to work unsocial hours, further research should analyze sectoral differences; for example, worker exposure to extreme time pressure has been evident in the female-dominated health- and social care sector during the COVID-19 pandemic and is expected to continue [56].

The analysis established an association between women’s health issues and shiftwork, thereby contributing to research that shows that adverse working time can be particularly detrimental to women’s health [57]. The negative health effects of shift work are known to be mediated by sleep restriction and circadian misalignment, but also by social misalignment [13]. This is particularly relevant for gender differences in work-life conflict. Women’s dual burden of employment and caring has been posited as health damaging with various studies finding associations between work–family conflict and physical ill health, depression, and hypertension [58]. Notwithstanding, Borgman et al. (2019) scoping review found that in Europe work-family conflict and health are linked, but longitudinal data do not always show robust causal interrelations [59]. Our analysis shows that time-based work-life conflict is particularly detrimental for women’s health bringing thereby further clarity to the issue.

Our analysis confirms that, from the health perspective, there is a need to further acknowledge the out-of-work temporal demands which compound on-the-job demands and to redesign the temporalities of working life. [60] There are several underlying reasons to the observed patterns of gender differences in working time-health associations. Inequality of opportunities between women and men in the labor market determines employment and working conditions that have consequences on health. Temporal aspects are at the core of the conditions, as they link directly to job quality and reflect further inequalities in time use.

The COVID-19 pandemic has revealed and exacerbated the health impacts of poor temporal job quality on women. A scoping review of work-related psychosocial risks in the female-dominated health and care sectors during the pandemic clearly showed how temporal elements of work impact health, including extreme time pressure, working unsocial hours, work overload in terms of patient numbers and hours, as well as additional and unintended shifts have reduced the autonomy of health workers to decide their time use. These factors have contributed to the catastrophic levels of stress and mental health disorders. In addition, the proximal stressors, such as lack of childcare and poor work-family balance have been a significant source of anxiety for the workers, in particular for female health workers and nurses. [56] Further, teleworking during the COVID-19 pandemic coupled with school/daycare closures has exacerbated the negative impacts of poor temporal job quality on women. Eurofound’s Living, Working, and COVID-19 surveys show that higher percentage of women than men reported difficulties in work-life balance, and the biggest increase between 2020 and 2021 among parents reporting they were too tired after work to do household tasks was found among women with young children, particularly women with young children who worked only from home. The lowest mental well-being in spring 2021 was registered among women aged 18–24 and women aged 35–44. [61,62] Structural gender inequalities are clearly observable in this situation; it was evident before and during the pandemic, that when care services are limited, women assume a greater share of unpaid care of children, older people, and people with disabilities [63,64,65,66].

Working part-time was associated in our study with a lower risk of health problems overall for women, which concurs with research that shows an immediate positive health effect for women with family caring responsibilities from working reduced flexible hours through the reduction of chronic stress [66]. However, part-time and temporary work are also associated with negative health impacts, mediated, for example, through job insecurity and the accumulation of economic disadvantage over the life-course [33,67]. Economic disadvantages are associated with higher rates of morbidity among women including chronic diseases and self-reported poor health [68]. We also found that women are more exposed to physical risk factors than men when working part-time, especially when this is involuntary. This finding highlights the importance of strengthening occupational safety measures in part-time and precarious employment. Further, involuntary part-time work can be a proxy for precarious employment [69,70], which impacts health through poor working conditions and socio-economic disadvantage [31,71,72]. While precarious employment is identified as a risk factor for poor mental health in general [73], women are more likely to work under precarious employment conditions [74,75]. Precarious employment has been found to contribute to women’s work-related mental health problems [3] through exposure to psychosocial risk factors, including high psychological demands, low control, lack of social support, and sexual harassment [76,77].

The following research and policy implications can be drawn from our analysis:-Further longitudinal and experimental research into the impact of working time on women’s health would be needed to substantiate the causal mechanism behind associations established in our analysis.-Gender segregation in the labor market is a persistent problem in Europe and warrants the monitoring of sectoral differences in temporal job quality.-There is a shift to a higher prevalence of remote and hybrid working practices in Europe. In this context, investigations into working time and conditions of home-based workers are highly relevant for research on determinants of health inequalities.-Life-course perspective should be adopted in the analysis of work-life conflict and its health effects. Gendered division of unpaid work may lead to accumulation of time-based strain from paid and unpaid work for women at particular life stages and increased health risks.-Precarious employment has been increasing in Europe over the past decades and the trend is expected to continue. This context merits further intersectional analyses on temporal job quality and health.-The exposure to work-related psychosocial risks should be prevented with gender sensitive occupational safety and health and working time regulations.-Work-hour mismatch (working more or less hours than desired) is gendered and linked to negative health outcomes. Working time polices should be further developed to address and balance the situation.-Working time regulation at the European Union level should be further developed to account for the different aspects of temporal job quality. Our study demonstrates that not only excessively long working hours should be addressed in regulation on health and safety grounds, but that other aspects of working time organizations, such as predictability, control, autonomy, and atypical schedules, have important health outcomes.

### Limitations

The results should be interpreted in view of the study’s limitations. First, job quality is a multidimensional concept, but we focus on the temporal aspects and do not explore the association between the health outcomes and other aspects of job quality or their interactions. Therefore, further research should explore the findings of our analysis in relation to the other dimensions of job quality, particularly within the framework of intersecting health inequalities. Second, the analysis of self-reported health does not provide objective clinical data on health status/outcomes. Individuals might perceive their health and risks differently, with a possible systematic difference in reports between men and women. We address this concern with our research design where health outcomes are compared within gender groups, and therefore any systematic differences in reporting health between men and women should not impact on the uncovered relationship between job quality and health outcomes analyzed by gender. Third, the analysis is based on cross-sectional data which puts limits on causal inferences. Reverse causality cannot be ruled out, with individuals in poor health having a constrained choice of employment and therefore self-selecting to jobs of poorer quality on temporal dimensions. The interpretation of the results and the explanations of possible causal mechanisms thus largely draw on the literature. Finally, some of the measures used in the EWCS do not allow distinguishing overall poor health from work-related poor health outcomes. Better comparative data that provide detailed information on job quality and health are needed. Overall, we address many of the identified limitations as well as potential measurement errors by carrying out and combining the analysis on two large scale and independent surveys, EWCS and LFS, although with limited individual level inferences.

## 5. Conclusions

Working time constitutes a structural determinant of health, with gender playing a pivotal role in this relationship [67]. The findings of our study confirm that it is important to continue to study factors of gender specific health outcomes from the social determinants of health perspective. Gender inequalities generated by differential exposure to working conditions will continue to evolve, and the ways in which time is perceived, acknowledged, valued, used, and assessed are central to job quality related health outcomes [78]. Quality jobs can have a significant impact in reducing negative health outcomes in both women and men.

Paradoxically, women continue to work more often part-time yet still suffer from ‘time poverty’ [79]; when paid working hours, time spent commuting to and from work and unpaid work time are all combined, the 2010 European Working Conditions Survey (EWCS) data found that women work, on average, 64 h a week compared to the 53 h worked by men [80]. Concurrently, data show that men more often work long hours in paid employment, which can have negative health impacts.

Temporal analysis of job quality can contribute to research on social determinants of health. The COVID-19 pandemic initiated home-based working at an unprecedented scale, with an expected permanent shift to more hybrid and teleworking modes. Evidence is solid on the work-life conflict mediated negative health impacts on women in this situation [81,82,83,84]. In addition, there has been an increase in time-related psychosocial risks (time pressure; long hours; lack of control and flexibility) for workers in the female-dominated essential sectors during the pandemic with immediate and long-term health effects [56]. Therefore, it is crucial that sociological and occupational health researchers continue to explore the role of gender in work environment and jobs [85].

Our findings strengthen the knowledge on how working time organization can impact workers’ health and wellbeing [10]—first, good temporal job quality is positively associated with subjective well-being of both women and men; and second, the health impacts of poor temporal job quality are particularly pronounced on women. Further research should explore the impact of temporal elements of job quality on women’s and men’s health independently by analyzing new data from the pandemic and post-pandemic time, as well as study the codependency of working time (paid and unpaid) between women and men within the context of the digital transformation and the related changes on how work is organized. As is characteristic to social determinants of health, they cannot be addressed only by the health sector, but in this case, via employment policies. Longitudinal studies on the impact of temporal elements of job quality on health can further track tends and should inform the development of employment policies that consider the health and gender equity perspective.

## Figures and Tables

**Figure 1 ijerph-19-04456-f001:**
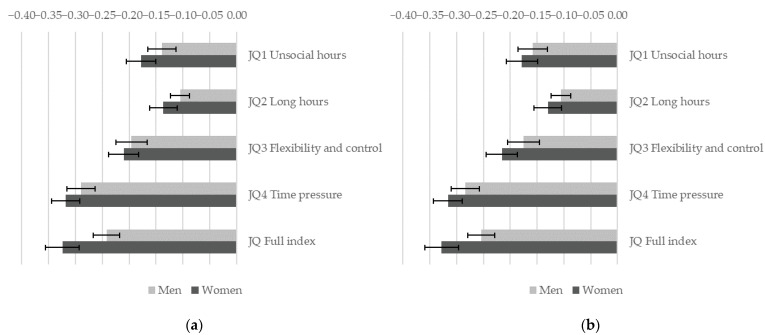
Gender differences in the association between job quality and a negative impact of work on health. A separate weighted logistic regression model is estimated for each indicator of job quality allowing for a gender-specific effect. Figure shows the estimated effect of job quality for men and women when job quality changes from a low value in the sample (10th percentile) to a high value (90th percentile) with 95% C.I. (**a**) Models control for education, age group and country; (**b**) models control for education, age group, country, sector, and occupation. EWCS 2015, EU-28 countries.

**Figure 2 ijerph-19-04456-f002:**
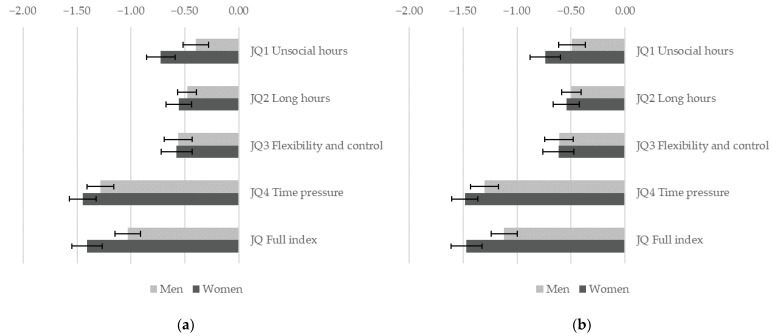
Gender differences in the association between job quality and health problems in the past 12 months. A separate weighted linear regression model is estimated for each indicator of job quality allowing for a gender-specific effect. Figure shows the estimated effect of job quality for men and women when job quality changes from a low value in the sample (10th percentile) to a high value (90th percentile) with 95% C.I. (**a**) Models control for education, age group and country; (**b**) models control for education, age group, country, sector, and occupation. EWCS 2015, EU-28 countries.

**Figure 3 ijerph-19-04456-f003:**
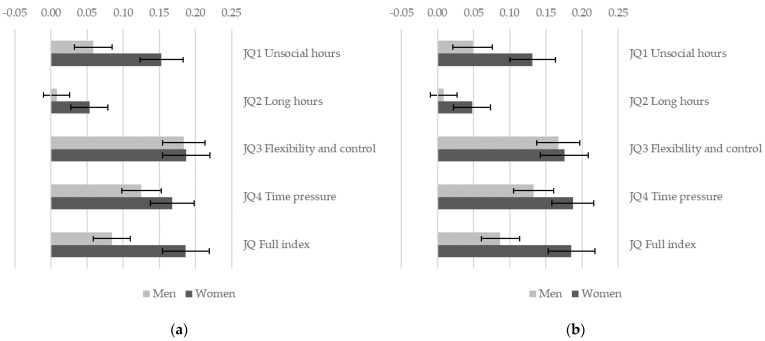
Gender differences in the association between job quality and sustainable work. A separate weighted logistic regression model is estimated for each indicator of job quality allowing for a gender-specific effect. Figure shows the estimated effect of job quality for men and women when job quality changes from a low value in the sample (10th percentile) to a high value (90th percentile) with 95% C.I. (**a**) Models control for education, age group and country; (**b**) models control for education, age group, country, sector, and occupation. EWCS 2015, EU-28 countries.

**Figure 4 ijerph-19-04456-f004:**
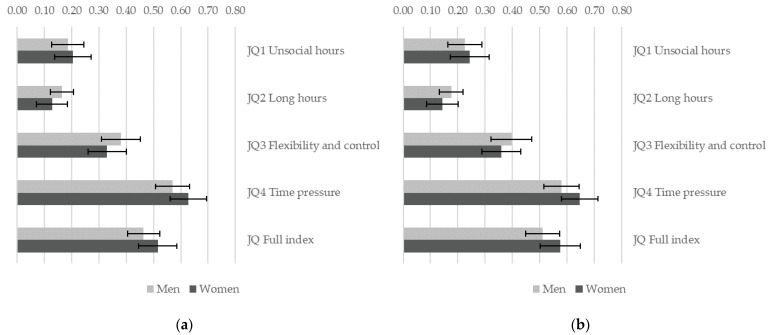
Gender differences in the association between job quality and subjective well-being. A separate weighted linear regression model is estimated for each indicator of job quality allowing for a gender-specific effect. Figure shows the estimated effect of job quality for men and women when job quality changes from a low value in the sample (10th percentile) to a high value (90th percentile) with 95% C.I. (**a**) Models control for education, age group and country; (**b**) models control for education, age group, country, sector, and occupation. EWCS 2015, EU-28 countries.

**Figure 5 ijerph-19-04456-f005:**
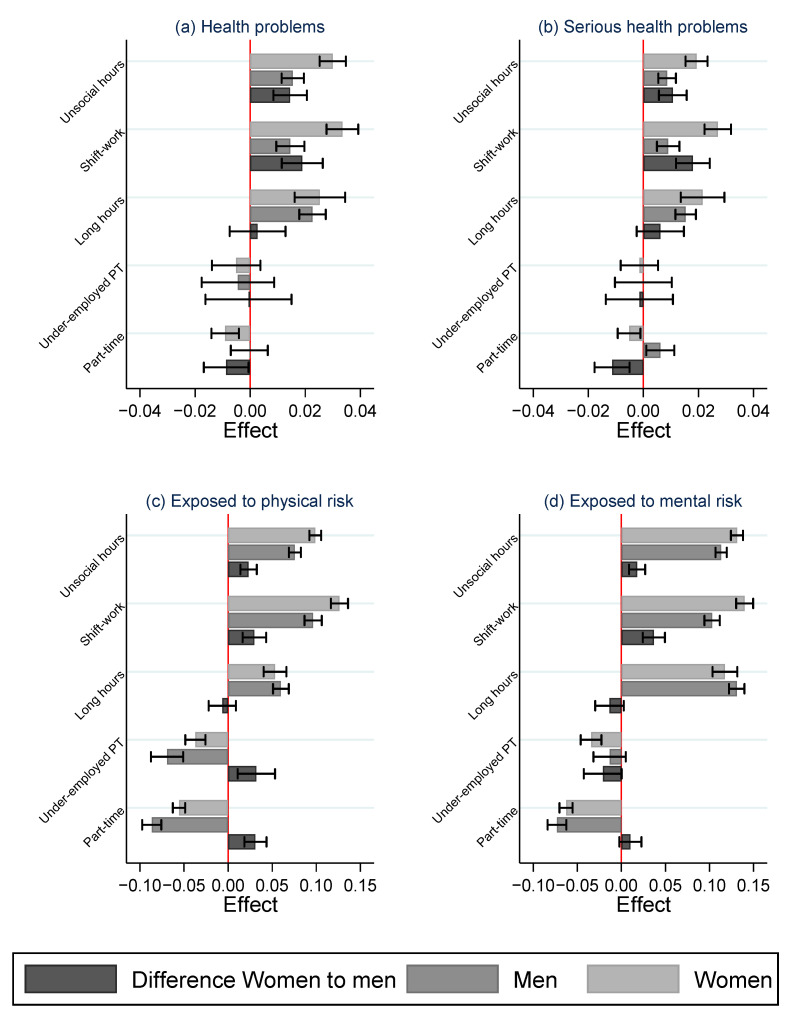
Gender differences in the association between job quality and a negative impact of work on health. A separate weighted logistic regression model is estimated for each indicator of job quality allowing for a gender-specific effect. Figure shows the estimated effect of job quality for men and women and the difference between them with 95% C.I. Models control for socio-demographic characteristics, occupation, industry and country (**a**) experience work-related health problems, (**b**) experience work-related health problems that limit daily activities, (**c**) being exposed to physical risks at the main job, and (**d**) being exposed to mental risks at work. LFS 2020.

**Figure 6 ijerph-19-04456-f006:**
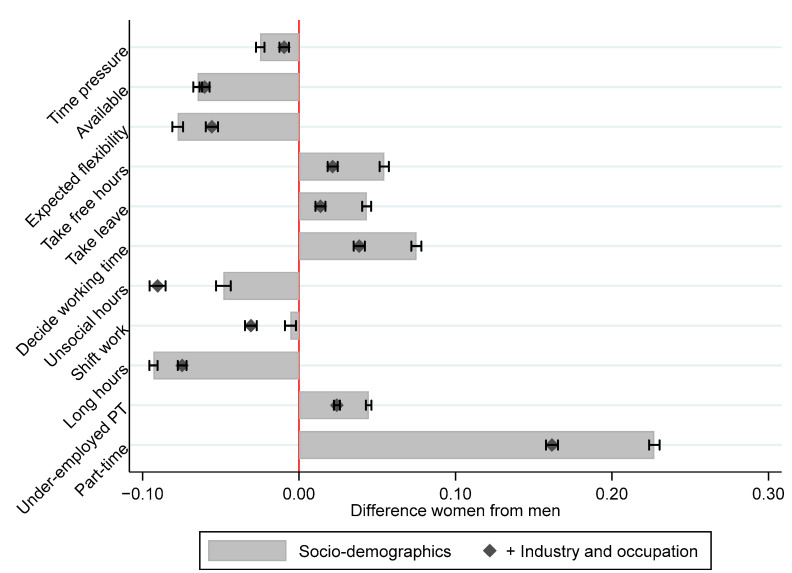
The difference of women compared to men (with 95% C.I.) for different indicators of temporal job quality, controlling for Socio-demographic characteristics and country fixed effects, and further adding industry by occupation fixed effects. A separate weighted regression is estimated for each outcome. Logistic regressions are run for each binary outcome, OLS otherwise. LFS 2019.

**Table 1 ijerph-19-04456-t001:** Temporal job quality indices: construction of measures and descriptive statistics.

Dimension of Temporal Job Quality	Items from the EWCS Used to Construct the Index	Mean (SD)	
JQ1 Unsocial hours	Work at night; work on Sundays; work on Saturdays; shift work *.	All Women Men	82.6 (20.2) 84.0 (19.6) 81.3 (20.7)
JQ2 Long hours	Work more than 48 h per week; work long days of more than 10 h; no recovery period (less than 11 h between two working days).	All Women Men	83.3 (26.0) 88.4 (21.6) 78.6 (28.8)
JQ3 Flexibility and control	Changes in work schedules **; short-term flexibility (taking an hour or two off for personal reasons); requested to come into work at short notice.	All Women Men	63.5 (20.0) 62.5 (19.6) 64.4 (20.2)
JQ4 Time pressure	Working to tight deadlines; factors constraining pace of work: colleagues, customer demands, production/performance targets, machine speed, boss; not having enough time to get the job done; working during free time to meet work demands.	All Women Men	67.4 (18.1) 69.1 (18.1) 65.8 (17.9)
JQ Full index	Overall temporal job quality: scale based on all items used in the construction of JQ1—JQ4.	All Women Men	67.7 (13.3) 69.8 (12.4) 65.8 (13.8)

All dimensions of job quality are normalized to 0–100 range with higher values indicating better job quality. * Contribution of shift work to index: no shift scores 100, permanent shifts scores 66, alternating shifts scores 33 and daily split shifts scores 0. ** The lowest score (0) is attributed to no worker control over employer-imposed changes to schedules, the highest score (100) is full control over one’s schedule.

**Table 2 ijerph-19-04456-t002:** Health and well-being: construction of measures and descriptive statistics.

Name of Measure	Construction of the Measure Based on the EWCS	Mean (SD)	
Negative impact of work on health	Subjective assessment of the impact of work on health: (1) affects mainly negatively, (0) affects mainly positively or does not affect.	All Women Men	0.25 (0.43) 0.23 (0.42) 0.27 (0.44)
Health problems in the past 12 months	Reported health problems experienced in the 12 months prior to the survey (min. 0, max. 10) *.	All Women Men	2.27 (2.06) 2.42 (2.09) 2.12 (2.03)
Sustainable work	Perceived ability to work in the current job or a similar one until the age of 60: (1) yes, (0) no.	All Women Men	0.72 (0.45) 0.70 (0.46) 0.74 (0.44)
Subjective well-being	Measured by the World Health Organization’s Well-Being Index (WHO-5) (min. 0, max. 5)	All Women Men	3.41 (1.00) 3.35 (1.02) 3.46 (0.97)

* Respondents could select from a list of 10 types of problems: hearing problems; skin problems; backache; muscular pains in shoulders, neck and/or upper limbs; muscular pains in lower limbs; headaches and eyestrain; injury; anxiety; overall fatigue; and other problems. The EWCS question does not explicitly link these problems to the job.

**Table 3 ijerph-19-04456-t003:** Description of data from the LFS survey.

Name of Measure	Description	Availability	Mean (SD)
Health problems	Person suffered any physical or mental health problems that were caused or made worse by work, apart from accidents.	2020 AHM	0.10 (0.31)
Serious health problems	Person experienced a health problem at work (as defined in Health problems) that limits the ability to carry out day to day activities either at work or outside work to some extent or considerably	2020 AHM	0.07 (0.26)
Exposed to physical health risk factors	Person is exposed at work to one of eleven risk factors that can affect physical health	2020 AHM	0.63 (0.48)
Exposed to mental well-being risk factors	Person is exposed at work to one of eight risk factors that can affect mental well-being.	2020 AHM	0.45 (0.50)
Part-time	Person works part-time rather than full-time	2000–2020	0.18 (0.39)
Long hours	Indicator: Person usually works more than 48 h per week.	2000–2020	0.09 (0.28)
Under-employed part-time	Person works part-time since they could not find a full-time job	2000–2020	0.04 (0.21)
Unsocial hours	Person usually works either evenings, nights, Saturdays, or Sundays	2000–2020	0.42 (0.49)
Shift work	Person works in shifts.	2000–2020	0.18 (0.38)
Free to take leave	Possibility to take one or two days of leave within three working days in the main job: 1 (very easy) to 4 (very difficult)	2019 AHM	0.47 (0.33)
Free to take hours off	Possibility to take one or two hours off in the main job for personal or family matters within one working day: from 1 (very easy) to 4 (very difficult)	2019 AHM	0.37 (0.34)
Decide working time	Who decides on working time: (1) worker fully decides; (2) worker with certain restrictions; (3) employer or organization	2019 AHM	1.58 (0.78)
Expected flexibility	Frequency to which the worker has to face unforeseen demands for changed working time in the main job: (1) less than every month or never; (2) less than every week but at least every month; (3) at least once a week	2019 AHM	1.62 (0.82)
Available	Worker was contacted during leisure time in the last two months to take action before the next working day for the main job: (1) not contacted in the last 2 months; (2) contacted on a few occasions; (3) contacted several times and not expected to act before the next working day; (4) contacted several times and expected to act before the next working day	2019 AHM	1.72 (1.01)
Time pressure	Frequency to which the person works under time pressure in the main job: (1) never, (2) sometimes, (3) often, (4) always	2019 AHM	2.31 (0.95)

Note: The table shows the weighted average (and standard deviation) of each variable in the sample. The sample size ranges from 440,944 (mental well-being) to 634,944 (health problems) in 2020 AHM; and from 468,161 (flexibility) to 473,618 (decide on working time) in 2019.

## Data Availability

Not applicable.

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
