# Peer review of "Temporal Dimensions of Job Quality and Gender: Exploring Differences in the Associations of Working Time and Health between Women and Men"

_ijerph, 2022, doi:10.3390/ijerph19084456_

Round 1
Reviewer 1 Report
This article is well written; the analyses are well motivated, and the issue is important. Overall, I don’t have too many comments on how the article is written or how models are chosen and performed. However, one important point took my attention. When going through the size of coefficients of association between different work arrangements and health, I noticed that they are really small. Also, it appears to me that gender differences in the figure 1 are really small, almost inexistent, as an example they are below 0.001 (similar in other figures). To sum up, I think that many coefficients are close to null effects, and that gender differences despite being significant (I assume that the sample is rather big, also more info on that would be useful) are actually not substantive. So, creating the whole story about the gendered effects and making an article out of it is perhaps not too justified.
My concrete suggestion would be that the authors carefully select variables with more substantial associations and exclude others, while mentioning that substantive differences between specific work dimensions/ genders are not there for those excluded. Please contextualize the size of coef in comparison to other similar studies.
Also, Figure 6 could be instead a starting figure that first descriptively shows the selection of women and men in different working arrangements.
Finally, I would appreciate the info on the distribution of the independent and dependent variables, and some major descriptive statistics as well as the info on the sample size for the datasets.
I would appreciate the use of correlational language: instead “ better job quality reduces” “ correlates”, instead of impact “association”.
Reviewer 2 Report
Research problem is important. Paper is well written, has a clear structure and easy to follows. The authors only could have emphasized even more their theoretical cotribution. Also, a bit more attention could have been paid to the impact of COVID pandemic.
Reviewer 3 Report
This article provides a sophisticated and comprehensive analysis of various indicators of working time-related job aspects and indicators of occupational well-being and health, differentiated by men and women, based on two publicly available representative European data sets. Overall, I appreciate the effort undertaken here and in my view the paper potentially makes an important contribution to the literature. Also, there should be no doubt that the article fits very well into the special issues on gender inequalities that it was submitted for.
That said, a major problem I see here, however, is a lack of provided statistical information. This pertains to the analyzed samples, formation and descriptive statistics of variables, and obtained results and effect sizes. My strong recommendation is to add this information into the manuscript to the fullest extent possible to increase its scientific utility and uptake.
Further the discussion part would profit form a more considerate elaboration of the possible mechanism behind the observed differential results pattern as well as influences of the pandemic situation. Finally, adding some more considered thoughts on practical implications seems warranted. I will elaborate these and some additional issues in my review below.
Title
I am not sure that the current title fully well reflects the content of the paper. I would recommend to change the second part to specify that differences in the associations between working time and health between women and men are the focus of this study, e.g., “Temporal dimensions of job quality and gender: Exploring differences in the associations of working time and health between women and men.”
Presentation of Studies
I believe it would be less confusing to present the two studies and their results separately and sequentially one after the other. Also, it be helpful if the two sets of analyses are consistently referred to and labelled as Study 1 and Study 2 throughout the manuscript.
Samples
Although these are secondary analyses of survey samples available for scientific use, nonetheless, it is important to specify the sample sizes (or range thereof) underlying the presented analyses for both studies. Also, some more information on the included control variables (especially sector, occupation, education age group) should be provided.
Indicators
Descriptive statistics of all independent and dependent variables should be provided. Possibly this information can be integrated into the respective Tables 1-3.
Further, given the purpose of the paper, it would also be relevant to differentiate these descriptive results by gender. This important additional information could be provided in an appendix.
Minor note: Extensively long information in the tables (e.g. Health problems in the past 12 months) should be put into a table note.
Minor note: Following study 1 (Table 1 and 2), Table 3 should be divided into two tables and reordered into independent (working-time related job quality) and dependent (health and well-being).
It is not completely clear how independent constructs in Study 1 were standardized to 0-100 scores. Please elaborate.
Results
At present, results are only displayed in graphs or figures, which is a problem as when subsequent research wants to refer to these results, the exact numbers are needed. Effect sizes, standard deviations and significance levels, therefore, need also be given in numbers, mostly these can probably be integrated into the figures (or be provided in an Appendix or as supplementary material).
Raw Correlations
Additionally, for further scientific use (e.g., meta-analysis) raw correlations for both studies would be desirable, possibly as an appendix or additional information. However, I regard this as optional, depending on the journal’s policies.
Discussion / Implications
The discussion section is still somewhat underdeveloped, especially with regard to the underlying reasons (conditions, causal mechanisms) of the observed patterns of gender differences in working time-health associations.
A second point that needs elaboration are the ramifications of the COVID pandemic for this study, which should be discussed in more detail (e.g., childcare responsibilities), where possible drawing on the available scientific literature and integrating results of relevant studies.
The limitations section is also still rudimentary and should be expanded, addressing, among others, limitations of cross-sectional, self-report data (reverse causality), lacking explanation of causal mechanisms etc.
Finally, at present, the study lacks a designated section on research and practical implications, which should be added. Especially, some practical implications from a trade union perspective seem warranted and could make an important contribution in rounding out the article.
Round 2
Reviewer 1 Report
The authors have managed to appropriately address the concerns. The value added of this study is in its comprehensiveness because it uses multiple datasets and measures.
The new title better describes what the article does as previous focus on women was not representative.
I recommend it for publication.
Reviewer 3 Report
I commend the authors for their thorough revision of the paper and implementation of suggested changes. In conjunction with the supplementary material, the article now provides all the requested statistical information. Also, the discussion section has been extended as recommended. Overall, I find all my comments addressed and am very pleased with the results. Again, I congratulate the authors on their important research.